# Reference Sequence Browser: An R application with a user-friendly GUI to rapidly query sequence databases

Sriram Ramesh[1], Samuel Rapp[2], Jorge Tapias Gomez[3]*, Benjamin Levine[4], Daniel Tapias-Gomez[5], Dickson Chung[1], Zia Truong[4]

1 Department of Computer Science and Engineering, University of California Santa Cruz, Santa Cruz, CA, United States of America, 2 Department of Ecology and Evolutionary Biology, University of California Santa Cruz, Santa Cruz, CA, United States of America, 3 Department of Computing and Information Science, Cornell University, Ithaca, NY, United States of America, 4 Department Biomolecular Engineering, University of California Santa Cruz, Santa Cruz, CA, United States of America, 5 Department of Cell and Molecular Biology, University of Texas Southwestern, Dallas, TX, United States of America

☯ These authors contributed equally to this work and are listed in alphabetical order.
* jt787@cornell.edu

**Data Availability Statement:** All data files are publicly available as it all comes from public databases, and in our publicly available GitHub we

## Abstract

Land managers, researchers, and regulators increasingly utilize environmental DNA (eDNA) techniques to monitor species richness, presence, and absence. In order to properly develop a biological assay for eDNA metabarcoding or quantitative PCR, scientists must be able to find not only reference sequences (previously identified sequences in a genomics database) that match their target taxa but also reference sequences that match non-target taxa. Determining which taxa have publicly available sequences in a time-efficient and accurate manner currently requires computational skills to search, manipulate, and parse multiple unconnected DNA sequence databases. Our team iteratively designed a Graphic User Interface (GUI) Shiny application called the Reference Sequence Browser (RSB) that provides users efficient and intuitive access to multiple genetic databases regardless of computer programming expertise. The application returns the number of publicly accessible barcode markers per organism in the NCBI Nucleotide, BOLD, or CALeDNA CRUX Metabarcoding Reference Databases. Depending on the database, we offer various search filters such as min and max sequence length or country of origin. Users can then download the FASTA/GenBank files from the RSB web tool, view statistics about the data, and explore results to determine details about the availability or absence of reference sequences.

## Introduction

Environmental DNA (eDNA) is an emerging field that has helped answer questions in several disciplines, including molecular ecology, environmental sciences, conservation biology, and paleontology [1, 2]. eDNA techniques are advantageous to traditional monitoring due to their non-invasive nature, ability to monitor aquatic communities, and the relative ease of training field workers.

have uploaded all the inputs necessary to re-create all the figures shown in the paper.

**Funding:** The team consists of undergraduate and graduate students and had no funding. However, over the course of the project, which spanned a few years, we participated in two competitions: UCSC NSF I-Corps Workshop and UCSC IDEA Hub. We were awarded $1,000 and $2,000, respectively. These funds mainly used to cover the publication fees for this paper.

**Competing interests:** We have no Competing Interests to report.

Environmental DNA workflows begin with acquiring samples from the environment under study so that genetic material can be extracted and then analyzed with various forms of Polymerase Chain Reaction (PCR) assays [2]. Depending on the specific goals of the study, eDNA primers are designed at varying levels of taxonomic inclusivity, often using either taxa-specific quantitative PCR (qPCR) or DNA metabarcoding PCR. Taxa-specific qPCR uses eDNA to detect a single targeted taxon, while eDNA metabarcoding aims to assess taxa richness.

Both qPCR assay design and taxonomic assignment in metabarcoding depend heavily on the "availability of DNA reference sequences in public data facilities (e.g., National Center for Biotechnical Information (NCBI), The Barcode of Life Data System (BOLD))" [3]. Reference sequences are pre-labeled sequences of DNA that allow researchers to either identify the unlabeled DNA they collect during their studies or design new species-specific primers for qPCR [4]. However, high-quality reference sequences are unavailable for many organisms, which has become a key factor limiting the broad application of eDNA techniques [5]. As such, screening genomics databases (e.g., NCBI Nucleotide and BOLD) or personalized reference sequence databases such as CRUX (Creating Reference Libraries Using eXisting tools) [6] for reference sequence availability and gaps is essential.

Additionally, detecting the absence of reference sequences is not the only challenge that eDNA scientists face. Through our own experience and interviews with over 60 scientists, we found that downloading the potentially thousands of sequences needed to create personalized reference databases is tedious for both new and experienced labs. Using the official websites for sequence databases, like NCBI Nucleotide or BOLD, to manually search for and download quality sequences can take days. Some researchers avoid this time-consuming process by writing scripts, but not all researchers possess those skills.

Completing both tasks in a time-efficient and accurate manner requires computational skills to search, manipulate, and parse multiple DNA sequence databases. Ecologists, however, often need more training in this area. For example, in California, most (74–80%) University of California and California State University undergraduates in ecological and environmental sciences had no formal training in programming skills in any language [7]. Bioinformatic assessments of large genetic databases can be challenging and time-consuming for classically trained ecologists and may result in bioinformatic work becoming a study expense and delaying the project.

To address these issues, we built the Reference Sequence Browser (RSB), a Graphical User Interface (GUI) tool that allows researchers to perform customized batch searches and downloads for reference sequences on the following publicly available databases: NCBI nucleotide, BOLD, and CALeDNA CRUX databases for metabarcoding [6, 8, 9]. Those interested in a simple but powerful way of viewing and downloading reference sequences from multiple genomics databases for one or multiple organisms and barcoding loci simultaneously will find that this tool saves time and provides insight into reference sequence availability and gaps. With the RSB, researchers can efficiently develop qPCR assays, determine the efficacy of particular barcoding markers to match an eDNA project's objective, and engage in more deliberate and specific sequencing efforts to address gaps in reference sequence availability.

## Related works

In recent years, numerous other tools have been developed to help scientists find sequence availability across multiple databases and create reference databases for their studies. However, there are significant differences between the services that these tools provide and what the RSB provides. Most tools that automate the process of searching public reference sequence databases either perform a sequence similarity search, like BOLDigger [10], gDAT [11], and

FROGS [12], or they instead perform in silico PCR, like rCRUX [13] and CRABS [14]. The RSB, by contrast, uses keyword search. While keyword search does open the tool up to vulnerabilities to inaccurate metadata, it comes with several advantages that complement the other search options. Firstly, keyword searches require significantly less computation time and the search parameters can be adjusted much more easily. These qualities are very useful when doing the iterative searching required to properly evaluate the sequence coverage that a public database offers. Additionally, depending on the details of the study, scientists may not have applicable primers to gather all of the reference sequences they need through in silico PCR.

While there are some tools that automate keyword searches, many of these tools require knowledge of programming languages that are less common in biology than R and often demand advanced command-line skills to fully utilize them [13]. Examples of such tools include Taxalogue [15] and CRABS. These skill requirements limit the accessibility of these tools for scientists and pose challenges for customizing the tools for individual needs. In contrast, the RSB was designed with ease of use in mind. It offers a browser version that requires no programming knowledge and features a simple, clean user interface. It is also entirely coded in R, facilitating modifications and personalizations by scientists in the field.

## Materials and methods

The protocol described in this peer-reviewed article is published on protocols.io (https://dx.doi.org/10.17504/protocols.io.q26g71zxqgwz/v1) and is included for printing purposes as S1 File. We highly recommend reading it, since it provides a thorough step-by-step guide on using the tool.

### Packages

The RSB is a Shiny GUI app [16], which is available online at https://sriramramesh.shinyapps.io/ReferenceSequenceBrowser/ or for download at https://github.com/SamuelLRapp/BlueWaltzBio. RSB uses the rentrez and bold CRAN packages to access the live NCBI and BOLD databases respectively and the Taxize package for some ease-of-use features. The app then visualizes the output using the shiny, ggplot, and plotly packages.

A complete list of our app's package dependencies can be found in "rsbPackages.xlsx" in the zip file above and this paper's "References" section.

### General features and workflow

The GUI is split into different tabs for each reference database, which the user can select by clicking on the respective tab at the top of the application window. Regardless of which database the user will be querying, the general workflow remains the same. Users will first be presented with a screen allowing them to upload a formatted CSV file with their query specifications (see our "S1 File" section to learn more about the CSV file template). While uploading a CSV file is optional, we highly recommend using this feature when doing large or expansive searches. Afterward, the RSB will present users with a screen where they may manually adjust their search parameters. After running a search, the app will display a table summarizing the results, and a list of download buttons will be present for the user to selectively download either the sequences or any metadata they will need for use with external bioinformatics tools.

For all databases, the user may choose to apply the taxize CRAN package's autocorrect feature to their list of organisms of interest. Instead of overriding the user's original input information, the tool will append any corrections to the user's list of organisms. For more

information about how taxize's corrections work, see the taxize CRAN package documentation [17].

Additionally, a tabular summary of the search results can be downloaded from any of the three database tabs, allowing users to view statistics about the overall quantity of data in each given reference database that matches their queries. These tables help users to quickly notice any missing information they may need for their study, and how over-represented or under-represented certain organisms or barcodes are in the search results. The rows of the summary table represent the various barcoding loci. The four columns of the table are the number of sequences found for this barcode, the percentage of all sequences found from the user's search that this barcode accounts for, the number of organisms that have at least one sequence for this barcode, and the number of organisms that have no sequences for this barcode. Fig 1 shows an example of a summary table.

While the summary tables are appropriate for seeing aggregated statistics about the search results of a user's query, the RSB also provides a more detailed view of a user's search results via the Coverage Matrix. The Coverage Matrix is a table that allows users to see how much of their needs can be "covered" by the data in the database they are querying. Each row of the table corresponds to an organism name, each column corresponds to a barcoding loci, and each cell displays the number of sequences found for the given organism-barcode pair. This layout allows the user to quickly scan for zeros or other low numbers in the table and understand where there are currently gaps in the sequence information publicly available to the eDNA community. After any search, users can download the Coverage Matrix (in future sections abbreviated as "CM") and the summary table. In addition to these two tables, the BOLD pipeline has a filter tab, which allows users to manipulate the results gathered by filtering the results by country of origin and giving the option to remove any entries also present in NCBI

## CRUX summary data

| Barcodes | Number of Sequences Found | Percent of Total Sequences Found | Number of Organisms with at Least one Sequence | Number of Organisms with no Sequences |
|---|---|---|---|---|
| Total | 5671 | 100 | 25 | 1 |
| 18S | 103 | 1.82 | 23 | 3 |
| 16S | 1334 | 23.52 | 14 | 12 |
| PITS | 0 | 0 | 0 | 26 |
| CO1 | 2946 | 51.95 | 23 | 3 |
| FITS | 1 | 0.02 | 1 | 25 |
| trnL | 0 | 0 | 0 | 26 |
| Vert12S | 1287 | 22.69 | 3 | 23 |

**Fig 1. Example of the CRUX summary table as shown in the RSB Shiny app.** All summary tables across databases have the same behavior and formatting. Their purpose is to summarize the data gathered and see the total number of species covered by each barcode.

Nucleotide. This allows users to avoid downloading duplicate FASTA files between BOLD and NCBI Nucleotide. It also has a few more tables and visualizations that allow the user to inspect the country origin of their data more closely. A lot more details on the specific components and tables of this app can be found in the S1 File protocol section of this paper.

## Use cases

We envision scientists using this tool to save time and money by speeding up the preprocessing step of checking the reference sequence coverage and downloading reference sequences needed for their studies. Using the many visualizations the tool provides, researchers can quickly determine whether the publicly available reference sequence coverage meets their needs. Below, we outline two specific use cases where the tool can speed up and aid environmental DNA studies. The examples below follow a hypothetical user who is interested in the invasive amphibians, fish, and invertebrates currently present in California. All species used for this hypothetical scenario are listed in Table 1 and were gathered from the California Department of Fish and Wildlife's website.

### Use case 1: Assessing online reference sequence availability

Knowledge of which organisms have publicly available reference sequences at known DNA barcoding loci is crucial for both metabarcoding studies and the design of new primers. In a metabarcoding study, taxa of interest can not be detected if there are no labeled reference sequences to compare to. Similarly, designing species-specific primers is only possible if there are enough reference sequences for both the target taxa and their phylogenetic relatives to represent the genetic diversity across individuals. Unfortunately, countless organisms are yet to be sequenced [5], so it is best practice to investigate the current state of sequence availability before beginning any study. The tool can easily accomplish this because it allows researchers to a) find how many reference sequences there are for each species per barcode and b) broaden or narrow their search depending on the results using the various filters and parameters.

While there are multiple factors to consider when choosing the right barcode for a study, an important one is to have sufficient coverage. Scientists can look at the RSB summary data table, which allows them to quickly compare barcodes against each other regarding how well they cover the organisms of interest. Directed by the summary data, scientists can then look at coverage for organisms at the most well-represented barcodes. Looking at the CM, the user can determine which organisms have enough sequences and which organisms currently need sequencing.

**Case 1.1: For metabarcoding studies.** It is important to recognize that while any of the three databases are suitable for this use case, the CRUX databases deserve special attention. These publicly available databases are specifically designed for metabarcoding studies and contain a curated list of high-quality barcodes. But, CRUX provides only a limited selection of searchable barcodes (CO1, 16S, 18S, PITS, FITS, trnL, Vert12S). So, if researchers need a

**Table 1. List of California invasive amphibians, fish, and invertebrates species by category.** We have also included this list as a CSV you can upload to the tool in your github in the file "invasive_CA_Species.csv".

| Category | Species |
|---|---|
| Amphibians | *Eleutherodactylus coqui, Lithobates catesbeianus, Xenopus laevis* |
| Fish | *Ctenopharyngodon idella* |
| Invertebrates | *Dreissena polymorpha, Dreissena bugensis, Eriocheir senensis, Euwallacea kuroshio, Euwallacea whitfordiodendrus, Pomacea canaliculata, Potamopyrgus antipodarum* |

## CRUX summary data

| | Barcodes | Number of Sequences Found | Percent of Total Sequences Found | Number of Organisms with at Least one Sequence | Number of Organisms with no Sequences |
|---|---|---|---|---|---|
| 1 | Total | 1059 | 100 | 6 | 5 |
| 2 | 18S | 6 | 0.57 | 3 | 8 |
| 3 | 16S | 62 | 5.85 | 6 | 5 |
| 4 | PITS | 0 | 0 | 0 | 11 |
| 5 | CO1 | 970 | 91.6 | 5 | 6 |
| 6 | FITS | 0 | 0 | 0 | 11 |
| 7 | trnL | 0 | 0 | 0 | 11 |
| 8 | Vert12S | 21 | 1.98 | 3 | 8 |

**Fig 2. Example of CRUX summary data table for the list of California invasive amphibians, fish, and invertebrates species.**

broader range of barcodes, they might start with CRUX and then use NCBI and/or BOLD to supplement their findings or bypass CRUX entirely. However, NCBI Nucleotide and BOLD also store lower-quality sequences and entries with incomplete metadata. Regardless of the chosen database, the primary value derives from the CM and Results Summary tables.

If the user is interested in determining the utility of the public CALeDNA CRUX databases for their study, they can start by following the CRUX pipeline until they reach the Summary Data Fig 2.

As shown in Fig 2, of the 11 invasive species searched, six organisms had at least one sequence, and 1059 sequences were found in total. 970 sequences were found in the CO1 barcode, and it includes five organisms. On the other hand, the 16S barcode captures all six detected organisms but only has 62 sequences in total. For the rest of this use case, let us suppose that the user is interested in using only the CO1 barcode for their study. To get more information, the user would then inspect a CM like Fig 3. It is important to note that for CRUX, users do not select the barcodes for the search. CRUX has a database for each of the following barcodes: 16S, 18S, CO1, FITS, PITS, trnL, and Vert12S. Our tool searches through all these databases and reports the results for each of these.

The CM allows you to quickly pinpoint very useful information. This information can range from the number of sequences found to highlighting missing data in a particular database. For example, in this particular case, a quick look reveals several key points:

- The CO1 barcode has coverage for five species: *Eleutherodactylus coqui, Xenopus laevis, Ctenopharyngodon idella, Pomacea canaliculata, and Potamopyrgus antipodarum*. Each of these species has a corresponding number representing the reference sequences found in CRUX. Furthermore, it becomes evident that Lithobates catesbeianus, Dreissena polymorpha, Dreissena bugensis, Eriocheir senensis, Euwallacea kuroshio, and Euwallacea whitfordiodendrus have no CO1 sequences in CRUX. For some species with no results, when the RSB searches at lower taxonomic resolutions, it found results at the genus and family levels.

## CRUX coverage matrix

| | 18S | 16S | PITS | CO1 | FITS | trnL | Vert12S |
|---|---|---|---|---|---|---|---|
| Eleutherodactylus coqui | order | 1 | phylum | 6 | phylum | 0 | 3 |
| Lithobates catesbeianus | family | family | phylum | family | phylum | 0 | family |
| Xenopus laevis | 3 | 14 | phylum | 18 | phylum | 0 | 11 |
| Ctenopharyngodon idella | order | 9 | order | 69 | order | 0 | 7 |
| Dreissena polymorpha | 2 | 4 | 0 | genus | phylum | 0 | class |
| Dreissena bugensis | genus | genus | 0 | genus | phylum | 0 | class |
| Eriocheir senensis | 0 | 0 | 0 | 0 | 0 | 0 | 0 |
| Euwallacea kuroshio | family | family | order | genus | order | 0 | family |
| Euwallacea whitfordiodendrus | 0 | 0 | 0 | 0 | 0 | 0 | 0 |
| Pomacea canaliculata | 1 | 4 | 0 | 520 | class | 0 | class |
| Potamopyrgus antipodarum | family | 30 | 0 | 357 | order | 0 | order |

**Fig 3. Example of CRUX CM table for the list of California invasive amphibians, fish, and invertebrates species.**

- Another important observation is that the row for Euwallacea whitfordiodendrus contains only zeros even though it should share similar results with Euwallacea kuroshio at the order, family, and genus levels. Because of this inconsistency, we decided to dig deeper and searched for Euwallacea whitfordiodendrus in the NCBI Taxonomy browser, and discovered that it is not registered in it. Therefore, when it was not found at the species level, it was not searched at broader scales, unlike the approach taken for Euwallacea kuroshio.

Considering these results, the user may either decide that using the 16S and CO1 primers alone will be sufficient for their study, or the user can broaden their search to the NCBI or BOLD search tabs and change any filters as needed. For example, Fig 4 displays the CM output from searching NCBI Nucleotide.

The NCBI search results in Fig 4 show that there are CO1 sequences for the species Lithobates catesbeianus, Dreissena bugensis, Dreissena polymorpha, and Euwallacea Kuroshio that were not present in CRUX. In addition, in contrast to the CRUX CM, NCBI allows users to choose the barcodes to be searched. However, for simplicity and consistency in this example use case, the same barcodes used in CRUX were applied. Additional searches could be made in BOLD. For more details, please refer to our protocol, which provides an in-depth explanation of the pipeline for each database.

**Case 1.2: For species-specific primer design.** In the case of species-specific primer design, the user would be interested in populating a local reference database with sequences for the target species, closely related species, and sympatric species [4]. In this case, we suggest looking at both the NCBI and BOLD tabs to get the full representation of the available barcodes for the organism(s) listed above.

Here, we describe the example of designing a primer for *Xenopus laevis*, one of the invasive amphibian species mentioned in case 1.1. Because the RSB BOLD tab allows users to deduplicate results also found in NCBI, users can start searching in either NCBI or BOLD first. To get a

## NCBI coverage matrix

| | 12S | trnL | ITS2 | CO1 + COI + COXI + COX1 | ITS1 | 16S | 18S |
|---|---|---|---|---|---|---|---|
| Eleutherodactylus coqui | 3 | 0 | 0 | 6 | 0 | 5 | 0 |
| Lithobates catesbeianus | 26 | 0 | 45 | 54 | 41 | 54 | 15 |
| Xenopus laevis | 70 | 0 | 38 | 342 | 34 | 714 | 577 |
| Ctenopharyngodon idella | 24 | 0 | 5 | 191 | 6 | 573 | 23 |
| Dreissena polymorpha | 5 | 0 | 16 | 125 | 5 | 904 | 25 |
| Dreissena bugensis | 0 | 0 | 0 | 39 | 0 | 22 | 2 |
| Eriocheir senensis | 0 | 0 | 0 | 0 | 0 | 0 | 0 |
| Euwallacea kuroshio | 0 | 0 | 1 | 8 | 4 | 0 | 1 |
| Euwallacea whitfordiodendrus | 0 | 0 | 0 | 0 | 0 | 0 | 0 |
| Pomacea canaliculata | 19 | 0 | 8 | 1796 | 1 | 215 | 52 |
| Potamopyrgus antipodarum | 36 | 0 | 99 | 403 | 95 | 78 | 104 |

**Fig 4. Example of NCBI CM table for the list of California invasive amphibians, fish, and invertebrates species.**

list of closely related species to search NCBI, users are advised to visit the NCBI taxonomy database to find the complete list of species names under a given taxonomic group. The user would first gather the highest quality sequences they can find from NCBI by using the most restrictive search parameters. For example, users should initially search NCBI by searching via the species and gene metadata fields, which is the default. The metadata for NCBI entries is not always complete, so searching in all fields is sometimes needed. By scanning the CM, the user can look for organisms with poor or no sequence coverage and loosen up the parameters if needed. Users can download the search statements used to search NCBI, which can be copy-pasted into NCBI's web interface to explore and validate results. Searching with [GENE] and [ORGN] but not using the taxize or sequence length parameters, NCBI produced the results shown in Fig 5.

Researchers can then quickly determine which barcodes have the most coverage in NCBI. In this case, the CO1 barcode group is the only one with any coverage. If more sequences are needed, the user can search in BOLD. One of the advantages of searching in BOLD is that researchers can simply search using a taxonomic group such as the genus *Xenopus*, and it will return results for all species under that group. Then, by excluding results found in NCBI shown in Fig 6, the user can observe the full unique set of available reference sequences across the two databases.

Following this, a user would search for results with a list of species that co-occur with the target (*Xenopus laevis*) in both BOLD and NCBI. The degree of sequence coverage of these co-species at any of 16S, 12S, or COI may help the user decide on a desirable loci for primer design.

### Use case 2: Creating a local reference sequence database

Regardless of whether a researcher is planning on conducting a metabarcoding study or species-specific primer design, creating a reference sequence database is required. In fact,

## NCBI summary data

| Barcodes | Number of Sequences Found | Percent of Total Sequences Found | Number of Organisms with at Least one Sequence | Number of Organisms with no Sequences |
|---|---|---|---|---|
| Total | 394 | 100 | 34 | 44 |
| 12S | 0 | 0 | 0 | 78 |
| trnL | 1 | 0.25 | 1 | 77 |
| ITS2 | 0 | 0 | 0 | 78 |
| CO1 + COI + COXI + COX1 | 393 | 99.75 | 34 | 44 |
| ITS1 | 0 | 0 | 0 | 78 |
| 16S | 0 | 0 | 0 | 78 |
| 18S | 0 | 0 | 0 | 78 |

**Fig 5. Example of NCBI summary data table for the genus Xenopus.**

designing a reference sequence database is listed as the first step of any quantitative PCR assay, regardless of what bioinformatic processing is done afterward, in "Development and Testing of Species-specific Quantitative PCR Assays for Environmental DNA Applications" [4]. The RSB facilitates this process by making downloading sequences in bulk from NCBI and BOLD simple, and the process for doing so follows directly from the steps outlined in case 1.2. For more step-by-step details on down loading sequences for each pipeline please refer to the protocol found in the S1 File section of this paper.

While the RSB can not output a fully formed reference database, we have made the output format generic, to ensure that the output is compatible with as many different pipelines or software suites as possible. Regardless of whether sequences are downloaded from the NCBI tab or the BOLD tab, fasta files will be downloaded as a zip file containing one file for each barcode loci included in the search results. Within each file, sequences are separated by newlines. This formatting is compatible with several alignment tools (e.g., Geneious, MegAlign Pro, DNADynamo), so scientists can move on to the next steps in the pipeline outlined in Klymus et al.'s publication.

## BOLD coverage matrix

| | COI-5P | CYTB | COXIII | COII | atp6 |
|---|---|---|---|---|---|
| Xenopus (Silurana) tropicalis | 1 | 0 | 0 | 0 | 0 |
| Xenopus laevis | 3 | 1 | 1 | 1 | 1 |
| Xenopus muelleri | 4 | 0 | 0 | 0 | 0 |

**Fig 6. Example of BOLD CM table for the genus Xenopus.**

## Discussion

Screening of and access to available reference sequences is necessary for the development of new quantitative PCR (qPCR) primers, to understand the efficacy of existing primers, and to understand gaps in what taxa can be identified in eDNA samples for given DNA barcodes. However, access to reference sequence information for the numerous organisms eDNA scientists hope to detect or exclude via primer design requires learning bioinformatic skills or investing large amounts of time into manual searches. With the RSB, users can easily screen NCBI, BOLD, and the public CALeDNA CRUX databases for reference sequence availability and download sequences for numerous organisms at multiple DNA barcodes. Therefore, users can generate deduplicated, up-to-date local sequence databases from the combined pools of BOLD and NCBI for primer design by using the NCBI filter in the BOLD tab. Additionally, summary statistics and other data visualization features allow users to glean other insights about reference sequence availability. Therefore, this tool can be used to explore gaps in reference sequence availability and help direct future sequencing efforts and funds. The RSB GUI bridges the gap between ecologists and computer scientists by providing efficient and intuitive access to NCBI, BOLD, and CRUX databases without requiring any programming.

While the RSB will be very helpful to eDNA scientists of all backgrounds and levels of experience, the tool does have limitations and space for some improvements. The RSB inherits the downsides of keyword searches, and although we have attempted to address as many as we could, the tool still has the following limitations:

- The scientific name searches performed by this tool are vulnerable to inaccurate results when searching for organism names for which homonyms exist. There is no homonym handling built into the tool, and adding some would make the tool more robust and easier to use. Currently, scientists would need to check the organism names they intend to search using a library like Taxize. Alternatively, most alignment software would throw errors when trying to process sequences with identical taxonomy labels but vastly different content, and these errors could notify users of the presence of homonyms in their dataset.

- In CRUX, when a species has no results, the tool attempts to search at lower taxonomic resolutions using the NCBI taxonomic backbone. This feature has two main limitations. Firstly, the phylogeny of a species may be updated, resulting in the tool searching up an unexpected taxonomy tree, for example if the genus is changed. In fact, this can be seen in the dataset used in Use Case 1.1, where *Lithobates catesbeianus* has been moved to the genus Aquarana. Another limitation is that the species genus itself may not be registered in the NCBI taxonomy backbone (also seen in Use Case 1.1 with *Euwallacea whitfordiodendrus*), in which case our tool will be unable to go up the taxonomic tree.

While it would be difficult to fix these limitations completely, automatically detecting such oddities in the data and giving the user more feedback would give users more control over the quality of their datasets. Additionally, new features, such as the ability to filter by geographic regions or to search for all species under a specific genus (or other higher taxonomic rank), could open up new use cases. A great quality-of-life feature would be highlighting all the changes made by Taxize so that users can more easily notice when Taxize adds unwanted organisms to the search query. Finally, adding a way to download sequences from the CRUX database similar to features NCBI and BOLD already possess would be a valuable improvement.

Our hope is that by opening up this tool to the eDNA community people can contribute to the github and add more quality of life features and/or incorporate more databases. One could also simply fork our repository and make their own personalized features that they would like

to have for their own use. Additionally, while we made it so the server can hold as many users as we could we have some scaling concerns, however, the tool can be easily used locally instead if any problems arise.

Lastly, our github contains all the necessary input files to replicate the results of this paper however it is important to keep in mind that these databases are constantly being updated and therefore the exact numbers in our tables may not exactly match searches done in the future.

## Supporting information

**S1 File. RSB protocol providing an in-depth step-by-step guide covering all tool features.** (PDF)

## Acknowledgments

We thank Hailey Nava for helping during the interview and research phase, Nitya Jain for helping in the BOLD pipeline particularly with the creation of the graphs, Jay Ferreira for helping us refactor and test our code, Dr. Rachel Meyer for guidance, review of the written process and helping us host our tool in the UCLA EEB servers.

Lastly, we would like to thank the over 60 scientists we have interviewed in the last 3 years who helped us identify workflow bottlenecks in the field of eDNA and improve this tool.

## Author Contributions

**Conceptualization:** Samuel Rapp, Benjamin Levine.

**Funding acquisition:** Samuel Rapp, Jorge Tapias Gomez, Benjamin Levine.

**Investigation:** Sriram Ramesh, Samuel Rapp, Jorge Tapias Gomez, Benjamin Levine.

**Project administration:** Sriram Ramesh, Samuel Rapp, Jorge Tapias Gomez.

**Software:** Sriram Ramesh, Jorge Tapias Gomez, Daniel Tapias-Gomez, Dickson Chung, Zia Truong.

**Supervision:** Sriram Ramesh.

**Validation:** Sriram Ramesh, Jorge Tapias Gomez.

**Visualization:** Jorge Tapias Gomez.

**Writing – original draft:** Sriram Ramesh, Samuel Rapp, Jorge Tapias Gomez, Daniel Tapias-Gomez.

**Writing – review & editing:** Sriram Ramesh, Samuel Rapp, Jorge Tapias Gomez, Benjamin Levine, Daniel Tapias-Gomez.

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
