## [Decision Letter · Decision Letter 0]

11 Jun 2024

PONE-D-24-11148Reference Sequence Browser: An R application with a User-Friendly GUI to rapidly query sequence databasesPLOS ONE

Dear Dr. Tapias Gomez,

Thank you for submitting your manuscript to PLOS ONE. After careful consideration, we feel that it has merit but does not fully meet PLOS ONE’s publication criteria as it currently stands. Therefore, we invite you to submit a revised version of the manuscript that addresses the points raised during the review process.

Please adjust the manuscript according to the comments of the two reviewers.

We look forward to receiving your revised manuscript.

Kind regards,

Arnar Palsson, Ph.D.

Academic Editor

PLOS ONE

Journal Requirements:

2.  Please assign your protocol a protocols.io DOI, if you have not already done so, and include the following line in the Materials and Methods section of your manuscript: “The protocol described in this peer-reviewed article is published on protocols.io (https://dx.doi.org/10.17504/protocols.io.[...]) and is included for printing purposes as S1 File.” You should also supply the DOI in the Protocols.io DOI field of the submission form when you submit your revision.

3. We note you have included a table to which you do not refer in the text of your manuscript. Please ensure that you refer to Table 1 in your text; if accepted, production will need this reference to link the reader to the Table.

**Additional Editor Comments:**

Please adjust the manuscript according to the comments of the two reviewers.

Reviewers' comments:

Reviewer's Responses to Questions

**Comments to the Author**

1. Does the manuscript report a protocol which is of utility to the research community and adds value to the published literature?

Reviewer #1: Yes

Reviewer #2: Yes

2. Has the protocol been described in sufficient detail?

To answer this question, please click the link to protocols.io in the Materials and Methods section of the manuscript (if a link has been provided) or consult the step-by-step protocol in the Supporting Information files.

The step-by-step protocol should contain sufficient detail for another researcher to be able to reproduce all experiments and analyses.

Reviewer #1: Partly

Reviewer #2: Yes

3. Does the protocol describe a validated method?

Reviewer #1: Yes

Reviewer #2: Yes

4. If the manuscript contains new data, have the authors made this data fully available?

Reviewer #1: Yes

Reviewer #2: N/A

**5. Is the article presented in an intelligible fashion and written in standard English?**

Reviewer #1: **No: **There are misalignments between text and figures that need correction. These are detailed below

Reviewer #2: Yes

6. Review Comments to the Author

Reviewer #1: This manuscript describes an R Shiny web application for supporting searches of sequence data repositories for the purpose of identifying reference sequence availability or discovery of gaps or missing sequences for target organisms. The app can be run online or the code downloaded to run locally. The Shiny app is designed to support searching 3 repositories for reference sequences for one to many organisms and standard barcode primers. The authors provide a reasonable justification for the application based on their experience and interviews with other scientists. The tool serves a niche for users who may not be savvy in accessing online data repositories to search for sequences. A key strength of the tool is that it provides a quick way to compare results across more than one data repository within the same platform rather than having to visit each data repository individually.

That said, there are some issues that could be improved. One is to review potentially related/similar work (e.g. crabs: Jeunen, G.-J., Dowle, E., Edgecombe, J., von Ammon, U., Gemmell, N. J., & Cross, H. (2023). CRABS—A software program to generate curated reference databases for metabarcoding sequencing data. Molecular Ecology Resources, 23, 725–738. https://doi.org/10.1111/1755-0998.13741

taxalogue: Noll NW, Scherber C, Schäffler L. taxalogue: a toolkit to create comprehensive CO1 reference databases. PeerJ. 2023 Dec 4;11:e16253. doi: 10.7717/peerj.16253. PMID: 38077427; PMCID: PMC10702336.) noting overlaps and differences and the degree to which the work fills a gap.

There are also some sections that could use some clarifications and corrections. In the section Case 1.1 under the third paragraph, the text states, ‘To get more information, the user would then inspect the CM like the Fig. 3.’ Figure 3 appears as a summary table rather than a coverage matrix (CM) as stated.

A little more careful alignment between text and figures would alleviate some potential confusions. For example, the text says,

‘As shown in the summary table, of the 11 invasive species searched, six organisms had at least one sequence.’ It is not clear how this information was derived from the summary table since it is not organized by organism. The text further says, ‘The vast majority of the sequences were found in the CO1 barcode and includes five organisms but the 16S barcode captures all six detected 148 organisms.’ That the majority are found in the CO1 barcode is clear from the summary table but one needs to refer to the coverage matrix (CM) to determine that there were 5 organisms detected under CO1 and six detected under 16S.

Similar additional clarification would help with the next paragraph. The text states, ‘one can see examples of the three different ways in which the RSB highlights missing information in the CRUX databases. Firstly, Lithobates catesbeianus, Dreissena polymorpha, and Euwallacea kuroshio are not in the database.

It would help to point out to readers the pattern that depicts this. From the Figure 2 example one can see that there are no sequences indicated for Lithobates catesbeianus (no numbers other than zero), but the text family and phylum appear so this makes sense.

Dreissena polymorpha does have sequences appearing for 18S and 16s so the above statement that there are no sequences for this organism appears incorrect. Perhaps this was meant to be Dreissena bugensis which does show no sequences appearing across the set of barcodes?

With respect to Figure 4. The text might be amended from this: ‘Here, there are CO1 sequences for species that were not present in CRUX. To ‘The NCBI search results show that there are CO1 sequences for species (Lithobates catesbeianus, Dreissena polymorpha, Dreissena bugensis and Euwallacea Kuroshio) that were not present in CRUX.

With respect to Figure 5, the caption is,’ Example of NCBI Summary Data table for the genus Xenopus’ However the figure title is ‘BOLD coverage matrix’. It appears that Figures 5 and 6 are inverted with respect to the captions /associated png files.

In summary the work has benefit for supporting search for and comparison of sequence availability across a set of data repositories. The authors however could provide some further discussion of related work (as pertinent) and provide more specifics and clarifications to assist readers in interpreting the tabular results.

Reviewer #2: In this paper, authors summarize an R application which allows users to access and download reference sequences from multiple genomic databases in order to assess reference sequence availability. It is clear that this application has a wide range of potential uses in metabarcoding studies, and that researchers of various levels of experience could benefit from using it. The compatibility with the taxize package and the ability to uncover gaps in information (e.g., species within genera of interest that possess or lack barcode data, underrepresented barcodes) are particularly useful features of the RSB that the metabarcoding and barcoding communities can greatly benefit from. I have very minor feedback for the authors:

1. Line 16: BOLD stands for Barcode of Life Data System; authors should make sure to use its correct name.

2. Lines 35-38: Example for the UC system is interesting but difficult to put into context. Is this expected to be high or low in comparison to other schools? Average? I realize that this is merely to illustrate that few students may have opportunities to gain experience in programming, but I wonder if the high-powered UC system actually represents an overestimate of how many students are exposed to this training. Maybe something like that could be mentioned.

3. Line 131: This paragraph is useful; I just worry that an inexperienced reader could interpret its message to say that a barcode should be chosen based on how many organisms it covers when a barcode should probably be chosen based on numerous factors (certainly inclusive of coverage, but also factors like species-level genetic variability). Maybe authors could mention something like that users can evaluate this factor “in addition to other factors specific to their study.”

4. Figures 1-3, 6: I was curious why trnL is included in these tables when it is a cpDNA marker and would not be used in exploring barcodes of amphibians. Upon further inspection, it seems that in the RSB itself the “barcodes” rows/columns appearing in these figures are just the standard list included in outputs. Does this mean, then, that coverage of only one cpDNA barcode can be explored for plants? If so, it may be worth a more detailed discussion of the utility of this tool for various types of organisms, or stating that this is not an exhaustive list of markers and researchers might be interested to use others not in the RSB (e.g., trnL has become the new standard for plants but those interested in a multi-marker approach or who might find utility for a more widely used marker like rbcL might benefit from knowing that they have other options). Since this tool is introduced as one that is beginner-friendly, additional information on why that set list of markers is used in the app and how individuals might consider alternatives (or expand on this framework to compare a standard barcode to lesser-used barcodes) could be helpful.

5. Line 200: Creating local reference libraries for study systems of interest is crucial to high-resolution and reliable metabarcoding data. I wonder if their importance could be more explicitly stated in this paragraph (the development of local reference libraries is also synergistic with the goals of RSB including identifying taxonomic and geographic gaps in barcode data availability).

6. Line 241: Could the authors briefly explain how users might deal with or anticipate homynyms in the framework of the RSB?

7. PLOS authors have the option to publish the peer review history of their article (what does this mean?). If published, this will include your full peer review and any attached files.

Reviewer #1: No

Reviewer #2: No

---

## [Author Response · Author response to Decision Letter 0]

24 Jul 2024

Dear Editors and Reviewers,

Thank you for the opportunity to revise our manuscript. We have carefully considered all the comments and suggestions provided by the reviewers and the academic editor. Responses to each comment are included in the Response_to_Reviewers.pdf file, as requested.

---

## [Decision Letter · Decision Letter 1]

19 Aug 2024

Reference Sequence Browser: An R application with a User-Friendly GUI to rapidly query sequence databases

PONE-D-24-11148R1

Dear Dr. Tapias Gomez,

We’re pleased to inform you that your manuscript has been judged scientifically suitable for publication and will be formally accepted for publication once it meets all outstanding technical requirements.

Kind regards,

Arnar Palsson, Ph.D.

Academic Editor

PLOS ONE

Additional Editor Comments (optional):

Reviewers' comments:

Reviewer's Responses to Questions

**Comments to the Author**

1. Does the manuscript report a protocol which is of utility to the research community and adds value to the published literature?

Reviewer #1: No

Reviewer #2: Yes

2. Has the protocol been described in sufficient detail?

To answer this question, please click the link to protocols.io in the Materials and Methods section of the manuscript (if a link has been provided) or consult the step-by-step protocol in the Supporting Information files.

The step-by-step protocol should contain sufficient detail for another researcher to be able to reproduce all experiments and analyses.

Reviewer #1: No

Reviewer #2: Yes

3. Does the protocol describe a validated method?

Reviewer #1: No

Reviewer #2: Yes

4. If the manuscript contains new data, have the authors made this data fully available?

Reviewer #1: Yes

Reviewer #2: Yes

**5. Is the article presented in an intelligible fashion and written in standard English?**

Reviewer #1: Yes

Reviewer #2: Yes

6. Review Comments to the Author

Reviewer #1: The issues identified in the first review have been satisfactorially addressed. There is an issue with the protocol. The link generates an error so that the protocol can not be accessed

Reviewer #2: The authors have addressed comments appropriately and I think the manuscript is suitable for publication.

7. PLOS authors have the option to publish the peer review history of their article (what does this mean?). If published, this will include your full peer review and any attached files.

Reviewer #1: No

Reviewer #2: No

---

## [Editor Report · Acceptance letter]

30 Sep 2024

PONE-D-24-11148R1 

PLOS ONE

Dear Dr. Tapias Gomez, 

I'm pleased to inform you that your manuscript has been deemed suitable for publication in PLOS ONE. Congratulations! Your manuscript is now being handed over to our production team.

Kind regards, 

on behalf of

Dr. Arnar Palsson 

Academic Editor

PLOS ONE